# DEMONSTRATION ACTOR CRITIC

## ABSTRACT

We study the problem of *Reinforcement learning from demonstrations (RLfD)*, where the learner is provided with both some expert demonstrations and reinforcement signals from the environment. One approach leverages demonstration data in a supervised manner, which is simple and direct, but can only provide supervision signal over those states seen in the demonstrations. Another approach uses demonstration data for reward shaping. By contrast, the latter approach can provide guidance on how to take actions, even for those states are not seen in the demonstrations. Specifically, such reward shaping approach trains an agent not only to imitate demonstrated actions when it encounters demonstrated states, but also to reach demonstrates states, when it confronts states that are not observed in the demonstration data. However, existing algorithms in the latter one adopt shaping reward which is not directly dependent on current policy, limiting the algorithms to treat demonstrated states the same as other states, and fail to directly exploit supervision signal in demonstration data. In this paper, we propose a novel objective function with policy-dependent shaping reward, so as to get the best of both worlds. We present a convergence proof for policy iteration of the proposed objective, under the tabular setting. Then we develop a new practical algorithm, termed as Demonstration Actor Critic (DAC). Experiments on a range of popular benchmark sparse-reward tasks shows that our DAC method obtains a significant performance gain over five strong and off-the-shelf baselines.

## 1 INTRODUCTION

Reinforcement Learning (RL) aims at solving sequential decision-making problems by learning through interacting with environments in a trail-and-error way. In many real scenarios, the existence of expert demonstrations has been well perceived as a critical value to enhance the capability of reinforcement learning algorithms. Recent years have witnessed many studies exploring the paradigm of learning from demonstration (LfD), which provides the learner with some demonstration data generated by expert policies. However, LfD yields a strong dependency on the assumption of demonstration optimality, which is usually inconsistent with the reality. To better integrate LfD with reinforcement learning, increasing efforts turn to reinforcement learning from demonstrations (RLfD), with a relaxation to the demonstration optimality assumption, which can lead to significantly boost sample efficiency of the RL process.

One major branch of RLfD proposes to leverage demonstration data in a supervised manner, by either using them to directly pretrain the policy (Silver et al., 2016) or supplement the learning target of the policy with a supervised objective when encountering the states in demonstration data (Rajeswaran et al., 2017a). Although appealingly simple and direct, such branch of RLfD unfortunately fails to fully exploit demonstration data as it can only provide supervision signal over those states observed in the demonstrations (Brys et al., 2015; Rajeswaran et al., 2017a; Reddy et al., 2019).

To deal with such problem, another major branch of RLfD takes advantage of the demonstrations in reward shaping, by either designing the demonstration-oriented potential-based reward shaping function (Brys et al., 2015; Sun et al., 2018), or inducing implicit dynamic reward shaping through learning a discriminator from demonstrations, which can distinguish between demonstrations and self-generated data (Zhu et al., 2018; Kang et al., 2018). These methods can provide guidance on how to take actions, even for those states are not seen in the demonstrations. Particularly, these methods train an RL agent not only to imitate demonstrated actions when it encounters the demonstrated states, but also to reach demonstrated states, when it confronts states that are not observed in the

demonstration data [1] (Ho & Ermon, 2016; Reddy et al., 2019; Wang et al., 2019). This is the core idea behind these reshaping reward based approaches. However, since the new adopted shaping reward yields no direct dependence on the current policy, this branch of methods, updating policy over demonstrated states in the same way as others by the reshaped value function, overlook the validity of such direct supervision for demonstrated states when learning the policy.

In order to provide both guidance for all states as well as direct supervision for demonstrated states, we propose a new objective function with policy-dependent shaping reward. To demonstrate the theoretic soundness of this approach, we first present a convergence proof for policy iteration of the proposed objective, under the tabular setting given the assumption of the existence of an expert policy $\pi_E(a|s)$. Furthermore, to cope with the problem of missing explicit expression of $\pi_E(a|s)$ in reality, we develop a new practical algorithm, called Demonstration Actor Critic (DAC), by making several approximations that can be implemented using deep neural networks. Intuitively, if the current state is not included in the demonstration, the agent will learn to update the policy merely relying on the reshaped Q-value function. Otherwise, the agent will take advantage of both expert information and the reshaped Q-value function to update the policy.

To demonstrate the effectiveness of our algorithm, we conduct experiments on the continuous physical locomotion tasks based on Mujoco (Todorov et al., 2012) in sparse-reward environments. In comparison with five strong and off-the-shelf baselines, the empirical results clearly show that our new DAC approach can attain consistent and significant improvements. Considering the recent concerns on reproducibility (Henderson et al., 2017), all of our reported results are based on experiments run across a large number of seeds.

The main contributions of this paper are summarized as:

- We introduce a novel RLfD objective with policy-dependent shaping reward, which provide both guidance for all states as well as direct supervision signal over demonstrated states.
- We derive a Demonstration Policy Iteration method with guaranteed convergence, under the tabular setting, by assuming the existence of the expert policy $\pi_E$..
- We develop Demonstration Actor Critic (DAC), a new practical algorithm to learn the policy for the continuous setting, given the missing expert policy in reality.
- We conduct empirical experiments in a couple of popular continuous tasks in sparse-reward environments to demonstrate the advantage of DAC as it consistently outperforms state-of-the-art baselines.

## 2 RELATED WORK

There is a growing interest in combining learning from demonstration (LfD) with reinforcement learning (RL). Recently there are three popular approaches under this problem setting: 1) utilizing demonstration data by adopting value-based RL algorithms; 2) leveraging demonstration data in a supervised manner; 3) using the demonstrations to reshape the original reward function.

For the first approach, they adopt value-based RL algorithms to utilize the demonstration data. Kim et al. (2013) proposes Approximate Policy Iteration with demonstrations (APID), which uses expert demonstrations to define linear constraints that guide the optimization of Approximate Policy Iteration (API). Piot et al. (2014) builds on a similar idea but integrate expert constraint directly into the minimization of the optimal Bellman residual (OBR). Following this line, Chemali & Lazaric (2015) bases on classification-based policy iteration and proposes Direct Policy Iteration with Demonstrations (DPID). More recently, thanks for the development of deep learning, DQfD (Todd et al., 2018) introduces LfD into DQN (Mnih et al., 2015), using the same additional structured classification loss than previous works. Besides, DQfD also adds demonstration data into the replay buffer in the same way as self-generated data. It employs a refined priority replay mechanism (Schaul et al., 2016) and assigns additional priority to demonstration data. However, these methods are limited by applications with discrete action spaces, due to the usage of max operator over the

---

[1]Although current state-action pair $(s_t, a_t)$ encountered by the agent may trigger low immediate reward $r(s_t, a_t)$ due to its unexposure in the demonstration data, the long-term reward $Q(s_t, a_t)$ is still likely to be high, especially if the agent can confront the demonstrated state-action pairs in later steps, so as to provides more reasonable guidance over state $s_t$.

whole action space. DDPGfD (Vecerik et al., 2017; Nair et al., 2017; Vecerik et al., 2019), which is built upon DDPG (Lillicrap et al., 2015), extends DQfD to continuous action domain. Moreover, NAC (Gao et al., 2018) uses a unified loss function to process both off-line demonstrations and on-line experience based on the maximum entropy reinforcement learning framework. Nonetheless, treating demonstration data in the same way as self-generated experience usually requires a tremendous number of high-quality demonstration, which are difficult to collect at scale, as discussed in (Kang et al., 2018).

For the second approach, they attempt to leverage the demonstration data in a supervised manner. For instance, Silver et al. (2016) proposes to pre-train the policy with the demonstration data as a policy initialization step for further reinforcement learning, and Rajeswaran et al. (2017b) augments the original policy loss with a behavior cloning loss during the policy training. Although appealingly simple and direct, such methods can only provide accurate supervision signal over those states that have been seen in the demonstrations.

For the third approach, they pursue to reshape the original reward function in order to align with the experience from the demonstrations. Specifically, Brys et al. (2015) introduces a reward reshaping mechanism by defining a heuristic potential function based on non-normalized multi-variate Gaussian. Besides, Sun et al. (2018) uses expert's value function as reward shaping, under the assumption of access to a reward-to-go oracle that provides an estimate of expert reward-to-go during training. Furthermore, Kang et al. (2018) introduces an implicit reward shaping via a parameterized discriminator, which aims to distinguish the demonstrated state-action pairs from self-generated pairs, and learn the policy with policy gradient methods. These methods can encourage the agent not only to imitate demonstrated actions, but also to visit demonstrated states. However, since the new adopted shaping reward yields no direct dependence on the current policy, this branch of methods pay rare attention to the validity of such direct supervision with respect to demonstrated states. To address this problem in the following of this paper, we develop an algorithm that can both provide guidance on all states and directly exploit the supervision signal on demonstrated states.

As one of the most popular IL algorithms, Generative Adversarial Imitation Learning (GAIL) (Ho & Ermon, 2016) trains a discriminator to distinguish whether a state-action pair is from the expert or the learned policy. Meanwhile, GAIL optimizes the policy by maximizing expected return with respect to the reward function, which is based on that discriminator. Though effective for imitation learning, GAIL cannot leverage the valuable reward signal given by the environment and may suffer from declining performance when the demonstration data is imperfect. By contrast, our algorithm can overcome such inherent limitation by introducing reward signals from the environment into the training process. Regarding the usage of discriminator, GAIL trains a discriminator to distinguish expert state-action pairs from other state-action pairs, while our method uses a discriminator to distinguish expert actions from other actions given expert states, which is totally different.

Some of previous works (Peters et al., 2010; Azar et al., 2012; Schulman et al., 2015; Haarnoja et al., 2017; Neu et al., 2017; Abdolmaleki et al., 2018; Haarnoja et al., 2018; Geist et al., 2019) studied the entropy regularization MDPs. Although there exist some similarities between our method and these works in terms of formulation, these methods are not very suitable for the RLfD problem studied in this paper. Particularly, these methods do not encourage the agent to reach demonstrated states (states visited by the expert strategy) explicitly, but it is a very important unique property of RLfD problem itself (Ho & Ermon, 2016; Kang et al., 2018; Reddy et al., 2019). By contrast, in our work, we delicately design a new policy-dependent shaping reward, in order to not only imitate demonstrated actions over these demonstrated states, but also reach demonstrated states, specifically for the RLfD problem. In addition, most of these works assume that the explicit expression of initial policy is available, but in our case, we can only access to expert demonstrations. To this end, we use the GAN technique to replace the necessary of explicit expression of $\pi_E$, and take advantage of support estimation techniques to estimate the indicator function of $\mathrm{supp}\,\pi_E(s)$, which also leads to an obvious difference between these entropy regularization works and our work.

## 3 BACKGROUND

### 3.1 MARKOV DECISION PROCESS

We consider the standard Markov Decision Process (MDP) (Sutton & Barto, 1998), defined by the tuple $\langle \mathcal{S}, \mathcal{A}, P, r, \gamma \rangle$, where $\mathcal{S}$ and $\mathcal{A}$ are the state space and the action space respectively, $P(s'|s, a)$ is the transition distribution, $r(s, a)$ is the reward function, and $\gamma \in (0, 1)$ is the discount factor.

Given a stochastic policy $\pi(a|s)$ that maps states to action probabilities, the performance of $\pi$ is usually evaluated by its expected discounted return $\eta(\pi)$:

$$\eta(\pi) = \mathbb{E}_{\tau \sim p_0, \pi, p}[\sum_{t=0}^{\infty} \gamma^t r(s_t, a_t)], \tag{1}$$

where $\tau = (s_0, a_0, s_1, ...)$ denotes a trajectory generated by policy $\pi$. Reinforcement Learning (RL) (Sutton & Barto, 1998) reflects the learning paradigm trying to infer a policy maximizing $\eta(\pi)$.

**Definition 1.** *(Occupancy measure). Let $\rho_\pi(s)$: $\mathcal{S} \to \mathbb{R}$ denote the unnormalized distribution of state visitation by following policy $\pi$ in the environment:*

$$\rho_\pi(s) = \sum_{t=0}^{\infty} \gamma^t Pr(s_t = s | \pi). \tag{2}$$

The unnormalized distribution of state-action pairs $\rho_\pi(s, a) = \rho_\pi(s)\pi(a|s)$ is called occupancy measure of policy $\pi$. Intuitively, the occupancy measure can be interpreted as the distribution of state-action pairs that an agent encounters when navigating the environment with policy $\pi$. An important property of the occupancy measure is the one-to-one correspondence with the policy, as described in the theorem 2 of (Syed et al., 2008).

### 3.2 DEMONSTRATION DATA SETTING

We formalize the demonstration data setting considered in this paper. The agent is provided with a few (and possibly imperfect) demonstrations as follows:

$$\mathcal{D}_E \triangleq \{(s_i, a_i)\}_{i=1}^{N} \overset{i.i.d.}{\sim} \rho_{\pi_E}(s, a).$$

$\mathcal{D}_E$ are sampled from executing an unknown expert policy $\pi_E$ in the environment. For the follow-up convergence guarantee, we have the following necessary assumption on the expert policy $\pi_E$:

**Assumption 1.** *The expert policy $\pi_E$ is a stochastic policy, and there exists a positive value $\delta$ satisfying that $\min_{a \in \mathcal{A}} \pi_E(a|s) \geq \delta$, $\forall s \in \mathcal{S}$.*

The point of this assumption is to ensure that $D_{KL}(\pi, \pi_E)$ is bounded by a constant $M$ for any $\pi \in \Pi$, under the tabular setting with $|\mathcal{A}| < \infty$.[2] Based on this, the augmented reward of our method (as shown later in Eq. 3) is also bounded, which can further lead to the convergence of demonstration policy evaluation (i.e. Lemma 1).

## 4 METHODOLOGY

In order to provide both guidance over all states and the supervision more directly on demonstrated states, we propose an objective function with policy dependent shaping reward:

$$\mathcal{J}(\pi) = \mathbb{E}_{(s,a) \sim \rho_\pi}[ \underbrace{r(s, a)}_{\text{extrinsic reward}} + \underbrace{\mathbb{1}_{s \in \text{supp}\, \rho_E(s)} \cdot \bar{D}_{KL}(\pi(\cdot|s), \pi_E(\cdot|s))}_{\text{policy-dependent KL augmented reward}}], \tag{3}$$

where $\mathbb{1}_{s \in \text{supp}\, \rho_E(s)}$ stands for the indicator function of $\text{supp}\, \rho_{\pi_E}(s)$[3], and $\bar{D}_{KL}(\pi(\cdot|s), \pi_E(\cdot|s)) \triangleq M - D_{KL}(\pi(\cdot|s), \pi_E(\cdot|s))$ [4]. More concretely, if state $s$ is unseen in the demonstrated states, the

---

[2]We refer readers to Appendix C.1 for a formal definition of the constant $M$.

[3]$\text{supp}\, \rho_{\pi_E}(s)$ denotes the support of state distribution of expert policy $\pi_E$. Intuitively, it represents the states seen in the demonstration data.

[4]This specific form of $\bar{D}_{KL}$ is to ensure that our augmented reward is always non-negative, where $M$ is the upper bound of $D_{KL}(\pi, \pi_E)$ as defined in Section 3.3.

augmented reward equals zero; Otherwise, the augmented reward is a positive number, indicating that a current policy $\pi$ closer to the expert policy $\pi_E$ will give rise to larger augmented reward. In this way, we can encourage the agent to both reach the demonstrated states and take action in a way similar to expert. Besides, we can prove that the optimal policy of our proposed objective is equal to that of the original RL objective, under the assumption that the expert policy $\pi_E$ is the optimal policy. The detailed proof can be found in Appendix B.

Since our shaping reward depends on the current policy $\pi$, optimizing the objective (Eq. 3) w.r.t the policy $\pi$ enables us to directly optimize the policy-dependent shaping reward itself. In other words, we can directly minimize the KL divergence between $\pi$ and $\pi_E$ over those demonstrated states. Detailed optimization will be illustrated in the policy improvement part in Eq. 8.

Inspired by soft value function in SAC (Haarnoja et al., 2018), we further introduce demonstration value function $V^\pi(s)$, by including the shaping reward at every time horizon:

$$V^\pi(s) = \mathbb{E}_\tau[\sum_{t=0}^{\infty} \gamma^t(r(s_t, a_t) + \mathbb{1}_{s \in \text{supp} \rho_E(s)} \cdot \bar{D}_{KL}(\pi(\cdot|s_t), \pi_E(\cdot|s_t))|s_0 = s]. \tag{4}$$

In a similar way, we also define demonstration Q-value function $Q^\pi(s, a)$ by including shaping reward at every time horizon, except the initial time horizon:

$$Q^\pi(s, a) = \mathbb{E}_\tau[\sum_{t=0}^{\infty} \gamma^t r(s_t, a_t) + \sum_{t=1}^{\infty} \gamma^t \mathbb{1}_{s \in \text{supp} \rho_E(s)} \cdot \bar{D}_{KL}(\pi(\cdot|s_t), \pi_E(\cdot|s_t))))|s_0 = s, a_0 = a]. \tag{5}$$

In the remainder of this section, we will first derive the Demonstration Policy Iteration method in Section 4.1, with the convergence guarantee under the tabular setting given the assumption of known $\pi_E(a|s)$. However, the explicit expression of $\pi_E$ is usually missing in reality. To tackle this challenge, we further develop the DAC algorithm in Section 4.2, which is more practical in real scenarios. Finally, we summarize the whole DAC algorithm in Algorithm 1.

## 4.1 DEMONSTRATION POLICY ITERATION

Given the assumption that $\pi_E(a|s)$ is known, we derive the demonstration policy iteration method, which alternates between policy evaluation and policy improvement. Our derivation is based on a tabular setting, for the purpose of theoretical analysis and convergence guarantee.

The policy evaluation step aims at computing the demonstration Q-value function of a policy $\pi$, which includes both the extrinsic reward and the shaping reward from demonstrations. Specifically, the demonstration Q-value function $Q^\pi(s, a)$ can be computed iteratively, starting from any function $Q : S \times A \to R$ and repeatedly applying a Bellman backup operator $\mathcal{T}^\pi$ given by:

$$\mathcal{T}^\pi Q(s_t, a_t) \triangleq r(s_t, a_t) + \gamma \mathbb{E}_{s_{t+1} \sim p(\cdot|s_t, a_t)}[V(s_{t+1})], \tag{6}$$

where

$$V(s_t) = \mathbb{E}_{a_t \sim \pi(\cdot|s_t)}[Q(s_t, a_t)] + \mathbb{1}_{s_t \in \text{supp} \rho_E(s)} \cdot (M - D_{KL}(\pi(\cdot|s_t), \pi_E(\cdot|s_t))). \tag{7}$$

The detailed evaluation process is formalized below.

**Lemma 1.** *(Demonstration Policy Evaluation). Consider the demonstration Bellman backup operator $\mathcal{T}^\pi$ in Eq. 6 and a initial Q function $Q^0 : \mathcal{S} \times \mathcal{A} \to \mathbb{R}$ with $|\mathcal{A}| < \infty$, and define $Q^{k+1} = \mathcal{T}^\pi Q^k$. Then the sequence $Q^k$ will converge to the demonstration Q-value of $\pi$ as $k \to \infty$.*

In the policy improvement step, for each state, we update the policy according to:

$$\pi'(\cdot|s) = \arg \max_{\pi' \in \Pi} \mathbb{E}_{a \sim \pi'(\cdot|s)}[Q^{\pi_{old}}(s, a)] - \mathbb{1}_{s \in \text{supp} \rho_E(s)} \cdot D_{KL}(\pi'(\cdot|s), \pi_E(\cdot|s)). \tag{8}$$

This particular choice of update can be guaranteed to result in an improved policy in terms of its demonstration Q-value function. This update rule consists of two different parts: the first one refers to the expectation of $Q^{\pi_{old}}(s, a)$, which encourages the agent to obtain more cumulative rewards and visit the demonstrated states, and the other part signifies the direct supervision signals over these demonstrated states. The indicator function of $\text{supp} \rho_E(s)$ determines whether the current state $s$ belongs to the demonstrated states, and if it is true, the KL divergence term will enforce the matching between the learned policy $\pi$ and the expert policy $\pi_E$.

We formalize the detailed improvement result in Lemma 2.

**Lemma 2.** *(Demonstration Policy Improvement). Let $\pi_{old} \in \Pi$ and let $\pi_{new}$ be the optimizer of the maximization problem defined in Eq. 8. Then $Q^{new}(s,a) \geq Q^{old}(s,a)$, $\forall(s,a) \in \mathcal{S} \times \mathcal{A}$ with $|\mathcal{A}| < \infty$.*

Overall, the complete demonstration policy iteration algorithm alternates between the policy evaluation and the policy improvement steps, and it will provably converge to the optimal policy, as demonstrated in Theorem 1. We refer readers to Appendix C for the detailed proof.

**Theorem 1.** *(Demonstration Policy Iteration). Repeated application of demonstration policy evaluation and demonstration policy improvement from any $\pi \in \Pi$ converges to a policy $\pi^*$ such that $Q^{\pi^*}(s,a) \geq Q^{\pi}(s,a), \forall\pi \in \Pi, \forall(s,a) \in \mathcal{S} \times \mathcal{A}$, assuming $|\mathcal{A}| < \infty$.*

## 4.2 DEMONSTRATION ACTOR CRITIC

The derived demonstration policy iteration above is presumed to be under the tabular setting with known explicit expression of the expert policy $\pi_E(a|s)$, which is, however, usually missing in reality. Thus, a critical challenge remains as how to develop a practical DAC algorithm under the common real scenarios where only demonstration data $\mathcal{D}_E$ exists.

First, we use function approximators (e.g. deep neural network), including value network $V_\varphi(s)$, Q-value network $Q_\phi(s,a)$ and policy network $\pi_\theta(a|s)$. Computing the ratio between the current policy $\pi(a|s)$ and the expert policy $\pi_E(a|s)$ is a necessary part of our derived theory, as shown in Eq. 7 and Eq. 8[5], especially when $\pi_E(a|s)$ is unknown. To tackle this challenge, we borrow the idea of the discriminative modeling in Generative Adversarial Networks (GANs) (Goodfellow et al., 2014), which is used to differentiate the real data from those created by the generator. In our case, we construct a discriminator network $D_w(s,a)$ that can distinguish whether an action is from $\pi(a|s)$ or $\pi_E(a|s)$ given state $s$. More formally,

**Theorem 2.** *Given the policy $\pi$ and expert policy $\pi_E$, we define that $D^*(s,a) \triangleq \arg\max_{D \in (0,1)} \mathbb{E}_{a \sim \pi}[\log D(s,a)] + \mathbb{E}_{a \sim \pi_E}[\log(1 - D(s,a))]$, then we have $\frac{\pi(a|s)}{\pi_E(a|s)} = \frac{D^*(s,a)}{1-D^*(s,a)}$.*

Based on this theorem, the ratio of $\pi(a|s)$ to $\pi_E(a|s)$ can be equivalently written as $\frac{D^*(s,a)}{1-D^*(s,a)}$. Then, we parameterize $D^*(s,a)$ to the discriminator network $D_w(s,a)$, and train the discriminator with the demonstration data. We continue deriving update rules for the value functions and policy. The value function $V_\varphi(s)$ is trained to minimize the squared residual error:

$$J_V(\varphi) = \mathbb{E}_{s \sim \mathcal{D}}[\frac{1}{2}(V_\varphi(s) - \mathbb{E}_{a \sim \pi_\theta}[Q_\phi(s,a) + \mathbb{1}_{s \in \text{supp}\,\rho_E(s)} \cdot (M - \log \frac{D_w(s,a)}{1 - D_w(s,a)})])^2]. \quad (9)$$

The Q-value function is trained to minimize the bellman residual:

$$J_Q(\phi) = \mathbb{E}_{s,a \sim \mathcal{D}}[\frac{1}{2}(Q_\phi(s,a) - \hat{Q}(s,a))^2], \quad (10)$$

where $\hat{Q}(s,a) = r(s,a) + \gamma\mathbb{E}_{s' \sim p(\cdot|s,a)}[V_\varphi(s')]$. Finally, the policy parameter can be learned by applying the policy update rule from Eq. 8:

$$J_\pi(\theta) = \mathbb{E}_{s \sim \mathcal{D}, a \sim \pi_\theta(\cdot|s)}\left[\mathbb{1}_{s \in \text{supp}\,\rho_E(s)} \cdot \log \frac{D_w(s,a)}{1 - D_w(s,a)} - Q_\phi(s,a)\right]. \quad (11)$$

We can find that the above objective includes both the Q-function $Q_\phi(s,a)$ and discriminator $D_w(s,a)$, which are represented by neural networks and can be differentiated. Hence, it is very convenient to apply the reparameterization trick, which can lead to a low-variance estimator. To this end, we reparameterize the policy using a neural network transformation:

$$a = f_\theta(\epsilon; s), \quad (12)$$

where $\epsilon$ is an input noise vector sampled from some fixed distribution, such as multivariate Gaussian. Then we can rewrite the objective in Eq. 11 as below:

$$J_\pi(\theta) = \mathbb{E}_{s \sim \mathcal{D}, \epsilon \sim \mathcal{N}}\left[\mathbb{1}_{s \in \text{supp}\,\rho_E(s)} \cdot \log \frac{D_w(s, f_\theta(\epsilon; s))}{1 - D_w(s, f_\theta(\epsilon; s))} - Q_\phi(s, f_\theta(\epsilon; s))\right]. \quad (13)$$

---

[5]The ratio is inside the KL divergence.

Similarly, the above learning objective w.r.t the policy network $\pi_\theta$ also consists of two different parts as Eq. 8. In particular, the gradient from the discriminator $D_w(s, a)$, which aims to distinguish whether an action is from the expert $\pi_E$ or the learned policy $\pi_\theta$ given the current state, will guide the agent to take actions in accordance with the expert when it encounters the demonstrated states. On the other hand, the gradient from the Q-value function $Q_\phi(s, a)$ will encourage the agent to obtain more cumulative rewards and explore the demonstrated region. Overall, our practical algorithm alternates between collecting experience from the environment, and updating the function approximators. We use off-policy data from a replay buffer to train the value and policy networks, and use demonstration data to train the discriminator network.

**Practical Expert Policy Support Estimation**    The indicator function of $\mathrm{supp}\,\rho_E$ is a key component in our DAC algorithm, which indicates whether current state $s$ belongs to demonstrated states. However, in practice, the expert policy is unknown and only a finite number of trajectories sampled according to $\pi_E$ are available. Consequently, we consider taking advantage of support estimation techniques to estimate this indicator function.

Recently, Wang et al. (2019) have established a connection between support estimation ideas and Random Network Distillation (RND) (Burda et al., 2018) - a method to design intrinsic reward for RL exploration based on the "novelty" of states visited. Their design of intrinsic reward is based on the observation that neural networks tend to have significantly lower prediction errors on examples similar to those on which they have been trained, which also inspires us to use prediction errors of networks trained on the demonstration states to approximate the indicator function of $\mathrm{supp}\,\rho_E$.

In particular, we introduce two neural networks: a label network representing the prediction task and a predictor network trained on demonstration states. Note that, the label network is randomly initialized but fixed then, and it takes a state as input with a scalar output, i.e., $f : \mathcal{S} \to \mathbb{R}$, and the predictor network $\hat{f} : \mathcal{S} \to \mathbb{R}$ is trained to minimize the expected MSE w.r.t its parameter $\psi$, as shown below:

$$\psi^* = \min_{\psi \in \Psi} \frac{1}{N} \sum_{i=1}^{N} ||\hat{f}(s_i; \psi) - f(s_i)||^2. \tag{14}$$

This process distills a randomly initialized neural network into a trained one. The prediction error is expected to be higher for the states that are outside demonstrated states. Based on that, the approximate indicator function can be eventually defined as follows:

$$\mathbb{1}_{s \in \mathrm{supp}\,\rho_E(s)} \approx \exp\left(-\mu ||\hat{f}(s; \psi^*) - f(s)||^2\right), \tag{15}$$

where $\mu$ stands for the temperature parameter. As the L2 norm is non-negative, the approximate indicator function ranges from 0 to 1. We choose $\mu$ to make that from demonstrated states are mostly close to 1.

Overall, the complete DAC algorithm can be summarized in **Algorithm 1**.

---

**Algorithm 1** Demonstration Actor Critic (DAC)

---

1: **Input:** Demonstration dataset $\mathcal{D}_E$, replay buffer $\mathcal{D}$, policy parameter $\theta$, demonstration value function parameter $\varphi$, demonstration Q-function parameter $\phi$, discriminator parameter $w$, label and predictor parameters $\psi_f, \psi_{\hat{f}}$.

2: Initialize the label parameter $\psi_f$, and train the predictor parameter $\psi_{\hat{f}}$ with dataset $\mathcal{D}_E$ via Eq. 14.

3: **for each iteration do**

4:     Sample trajectories by using the policy network $\pi_\theta$ and store transitions into $\mathcal{D}$.

5:     **for** $k = 1, .., K$ **do**

6:         Sample batch from $\mathcal{D}_E$ and update the discriminator parameter $w$ via Theorem 2.

7:         Sample batch from $\mathcal{D}$, update the policy parameter $\theta$ via Eq. 13, the demonstration

8:         value function parameter $\varphi$ via Eq. 9, the demonstration Q-value function parameter $\phi$

9:         via Eq. 10.

10:     **end for**

11: **end for**

---

## 5 EXPERIMENTS

For the experiments below, we provide empirical results to answer the following questions:

1. Can our DAC algorithm achieve better performance than other counterparts, from the same RLfD setting or other settings?
2. What is the key ingredient in our algorithm that introduces better empirical results?

To answer the first question, we evaluate our method against several baselines on five sparse physics-based based control benchmarks. Regarding the second question, we explore ablation analysis of the two major components in our algorithm (namely the KL shaped reward and direct KL policy loss). Due to the space limit, we defer more detailed specifications into the appendix.

### 5.1 COMPARATIVE EVALUATION

**Experiment Settings** We conduct experiments on the sparse version of five popular continuous control tasks (*Hopper-v1, HalfCheetah-v1, Walker2d-v1, Ant-v1, Humanoid-v1*) from OpenAI Gym (Duan et al., 2016) . Specifically, we use the delayed version of the Mujoco domains[6] as (Zheng et al., 2018; Oh et al., 2018) did, where the reward is made sparse by accumulating the reward for $N = 10$ timesteps before it to the agent. Expert's trajectories were collected from the expert policy released by the authors of the original GAIL[7]. In particular, the maximum number of expert trajectories was chosen as (Ho & Ermon, 2016; Jeon et al., 2018), i.e. 240 for *Humanoid-v1* and 25 for all other tasks. For all tasks, feedforward neural networks with two hidden layers are used to represent the policy and value functions, where 256 hidden units for each hidden layer and relu activations are used. For the policy, Gaussian policy is used with both mean and variance dependent on the state. During training, we use the Adam optimizer (Kingma & Ba, 2015), with a learning rate of $3 \times 10^{-4}$ for all networks and set $K = 1$ to make the algorithm faster in terms of wall clock time. We refer readers to Appendix D for more implementation details.

For comparative evaluation, We compare our DAC algorithm against five strong and off-the-shelf baselines including:

1. Policy Optimization with Demonstration (POfD): the algorithm of (Kang et al., 2018) leveraging demonstration to reshape the reward function.
2. Policy Optimization with Demonstration with Behavior Cloning (POfDBC): the simple combination of POfD algorithm and an augmented behavior cloning loss.
3. Deep Deterministic Policy Gradient from Demonstration (DDPGfD): the algorithm of Vecerik et al. (2017) putting demonstrations into the replay buffer as self-generated data.
4. Generative Adversarial Imitation Learning (GAIL): the algorithm of (Ho & Ermon, 2016), a popular imitation learning method, mimicking the expert behaviour by matching the occupancy measure between the expert policy and the learned policy.
5. Soft Actor Critic (SAC): the algorithm of (Haarnoja et al., 2018), a state-of-the-art off-policy reinforcement learning method, building upon the maximum entropy reinforcement learning framework.

We report the average of the score of the agent over 10 episodes for every 10k steps performed in the environment, as shown in Fig. 1. The results show that DAC performs consistently across all tasks, and outperforms all strong baselines in terms of both sample efficiency and final performance. Besides, on most benchmarks, DAC displays smaller shaded region than other baselines, which implies that DAC can be more stable and robust across different random seeds. Observing the learning curves of different methods, it is clear that SAC cannot learn very fast without the help of expert demonstrations, especially when the feedback is sparse. On the other hand, GAIL can improve quickly in the early stage of training process of several tasks, e.g. *Walker2d-v1*, but it tends to be limited by the quality of demonstration later. Under the same RLfD setting, DAC also significantly outperforms other counterparts, such as POfD, DDPGfD, and POfDBC, which is a simple combination of POfD

---

[6]Publicly available in https://github.com/Hwhitetooth/lirpg
[7]https://github.com/openai/imitation

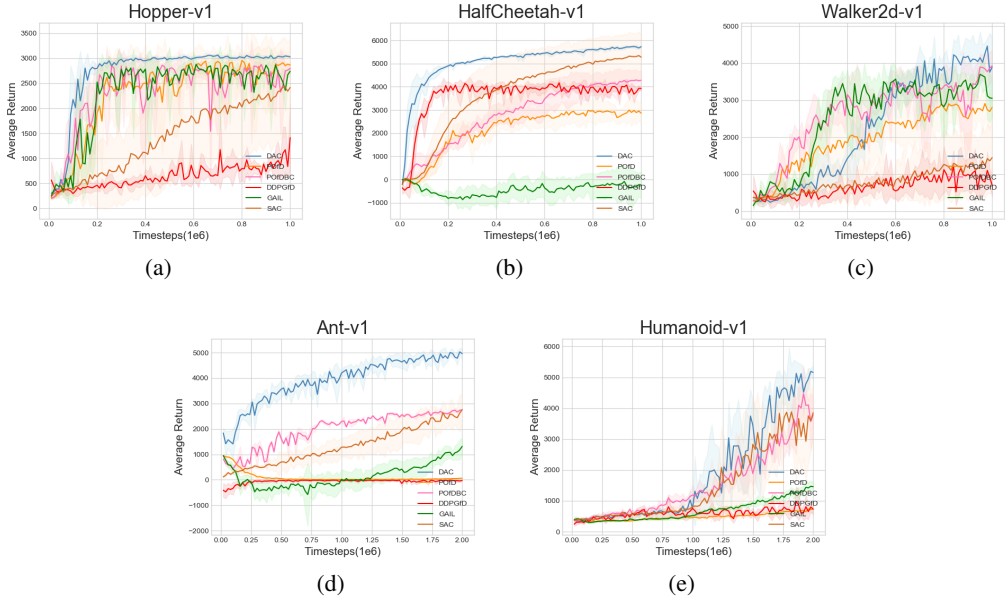

Figure 1: Learning curves of DAC and five baselines on sparse continuous control benchmarks. Solid curves depict the mean of ten trials and shaded regions correspond to standard deviation among trials. DAC (blue) performs consistently across all tasks and outperforms all strong baselines.

algorithm and an augmented behavior cloning loss, across all the benchmarks. Furthermore, we can see that the more complex benchmarks, e.g. *Ant-v1* and *Humanoid-v1*, are exceptionally difficult to be solved by other baselines. In stark contrast, DAC can learn the policy fast and steadily.

## 5.2 ABLATION STUDY

The previous results suggest that our proposed method can outperform other strong baselines on several challenging tasks. Now we will further perform ablation study to investigate the influence of two major components inside our algorithm, i.e., the KL reward shaping and direct KL policy loss, on the overall performance of our DAC.

We conduct two ablation experiments on *HalfCheetah-v1* by removing the KL augmented reward when computing Q-function, and the direct KL policy loss during policy improvement, respectively[8]. The comparative results are shown in Fig. 2.

We can observe that removing either of the two components will lead to the obvious degradation in learning performance. This suggests that both the KL reward shaping and the direct KL policy loss effectively contribute to the overall performance of our DAC algorithm. Furthermore, we find the degradation of removing the direct KL policy loss is even larger than that of removing the KL reward shaping in this *HalfCheetah-v1* task, which well demonstrates that the exploitation of direct supervision signal present in demonstration data may play an important role for better learning performance.

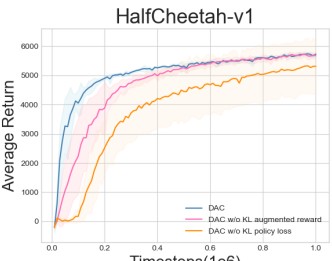

Figure 2: Ablation curves.

---

[8]These two components correspond to two different parts in Eq. 8, respectively.

## 6 CONCLUSIONS AND FUTURE WORK

In this paper, we studied reinforcement learning from demonstration (RLfD) and focused on developing a novel method that can not only provide guidance on all states, but also pass supervision signal more directly on demonstrated states. We propose a novel objective function with policy-dependent shaping reward, and derive both theoretical guarantee (Demonstration Policy Interation) and practical algorithm (Demonstration Actor Critic) for our objective function. Experiments on a range of popular benchmark sparse-reward tasks show that our method consistently achieves much higher performance than several strong and off-the-shelf baselines. For future work, we will explore the direction for improving the robustness of DAC in terms of the demonstration quality, which is not particularly modeled in current algorithm.

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

## A  DEFINITIONS

The definition of our proposed new objective is that:

$$\mathcal{J}(\pi) = \mathbb{E}_{(s,a)\sim\rho_\pi}[r(s,a) + \mathbb{1}_{s\in\text{supp}\,\rho_E(s)} \cdot (M - D_{KL}(\pi(\cdot|s), \pi_E(\cdot|s)))]. \tag{16}$$

And it also has another equivalent form as follows:

$$\mathcal{J}(\pi) = \mathbb{E}_\tau[\sum_{t=0}^\infty \gamma^t(r(s_t, a_t) + \mathbb{1}_{s_t \in \text{supp}\,\rho_E(s)} \cdot (M - D_{KL}(\pi(\cdot|s_t), \pi_E(\cdot|s_t))))]. \quad (17)$$

Based on the above objective, we further introduce demonstration value function $V^\pi(s)$, by including the shaping reward at every time horizon:

$$V^\pi(s) = \mathbb{E}_\tau[\sum_{t=0}^\infty \gamma^t(r(s_t, a_t) + \mathbb{1}_{s_t \in \text{supp}\,\rho_E(s)} \cdot (M - D_{KL}(\pi(\cdot|s_t), \pi_E(\cdot|s_t))))|s_0 = s]. \quad (18)$$

Similarly, we also define demonstration Q-value function $Q^\pi(s, a)$ by including the shaping reward at every time horizon, except the first time horizon:

$$Q^\pi(s, a) = \mathbb{E}_\tau[\sum_{t=0}^\infty \gamma^t r(s_t, a_t) + \sum_{t=1}^\infty \gamma^t \mathbb{1}_{s_t \in \text{supp}\,\rho_E(s)} \cdot (M - D_{KL}(\pi(\cdot|s_t), \pi_E(\cdot|s_t)))|s_0 = s, a_0 = a]. \quad (19)$$

With these definitions, $V^\pi(s)$ and $Q^\pi(s, a)$ are connected by:

$$V^\pi(s) = \mathbb{E}_{a \sim \pi(\cdot|s)}[Q^\pi(s, a)] + \mathbb{1}_{s \in \text{supp}\,\rho_E(s)} \cdot (M - D_{KL}(\pi(\cdot|s), \pi_E(\cdot|s))). \quad (20)$$

$$Q^\pi(s, a) = r(s, a) + \gamma \cdot \mathbb{E}_{s' \sim p(\cdot|s,a)}[V^\pi(s')]. \quad (21)$$

# B ON THE OPTIMAL POLICY INVARIANCE OF OUR OBJECTIVE FUNCTION

Recall that our proposed objective is defined as:

$$\mathcal{J}(\pi) = \mathbb{E}_{(s,a) \sim \rho_\pi}[r(s, a) + \mathbb{1}_{s \in \text{supp}\,\rho_E(s)} \cdot (M - D_{KL}(\pi(\cdot|s), \pi_E(\cdot|s)))]. \quad (22)$$

*Proof.* Assume that $\pi^* = \arg\max_{\pi \in \Pi} \mathbb{E}_{(s,a) \sim \rho_\pi}[r(s, a)]$ (in other words, $\pi^*$ represents the optimal policy of original RL objective), and the expert policy is perfect: $\pi_E = \pi^*$. We prove that $\pi^*$ is also the optimal policy of our proposed objective $\mathcal{J}(\pi)$ as follows:

$$
\begin{aligned}
\mathcal{J}(\pi^*) &= \mathbb{E}_{(s,a) \sim \rho_{\pi^*}}[r(s, a) + \mathbb{1}_{s \in \text{supp}\,\rho_{\pi^*}(s)} \cdot (M - D_{KL}(\pi^*(\cdot|s), \pi^*(\cdot|s)))] \\
&= \mathbb{E}_{(s,a) \sim \rho_{\pi^*}}[r(s, a) + \mathbb{1}_{s \in \text{supp}\,\rho_{\pi^*}(s)} \cdot M] \\
&= \mathbb{E}_{(s,a) \sim \rho_{\pi^*}}[r(s, a)] + \mathbb{E}_{(s,a) \sim \rho_{\pi^*}}[\mathbb{1}_{s \in \text{supp}\,\rho_{\pi^*}(s)} \cdot M] \\
&= \mathbb{E}_{(s,a) \sim \rho_{\pi^*}}[r(s, a)] + M \\
&\geq \mathbb{E}_{(s,a) \sim \rho_\pi}[r(s, a)] + \mathbb{E}_{(s,a) \sim \rho_\pi}[\mathbb{1}_{s \in \text{supp}\,\rho_{\pi^*}(s)} \cdot (M - D_{KL}(\pi(\cdot|s), \pi^*(\cdot|s)))], \forall \pi \in \Pi \\
&= \mathcal{J}(\pi), \forall \pi \in \Pi.
\end{aligned}
$$
$$(23)$$

Therefore, we have that $\pi^*$ is also the optimal policy of our proposed objective. This implies that if a policy $\pi$ is the optimal policy of original RL objective, then it is also the optimal policy of our proposed objective.

Next, we will continue to prove the inverse proposition: if a policy $\pi$ is the optimal policy of our proposed objective function, then it is also the optimal policy of original RL objective. Assume that $\pi^\#$ is the optimal policy of our proposed policy, and $\pi^*$ still denotes the optimal policy of original RL objective. Due to the optimality of $\pi^\#$ in terms of $\mathcal{J}$, we have $\mathcal{J}(\pi^\#) \geq \mathcal{J}(\pi), \forall \pi \in \Pi$. In particular, $\pi^*$ is a specific policy $\pi \in \Pi$. Therefore, we have $\mathcal{J}(\pi^\#) \geq \mathcal{J}(\pi^*)$. From the above Eq. 23, we get that $\mathcal{J}(\pi^*) = \mathbb{E}_{(s,a) \sim \rho_{\pi^*}}[r(s, a)] + M$, so $\mathcal{J}(\pi^\#) \geq \mathbb{E}_{(s,a) \sim \rho_{\pi^*}}[r(s, a)] + M$. However, we can easily find that the upper bound of $\mathcal{J}(\pi)$ is also $\mathbb{E}_{(s,a) \sim \rho_{\pi^*}}[r(s, a)] + M, \forall \pi \in \Pi$, according to Eq. 22. Here we come:

$$\mathbb{E}_{(s,a) \sim \rho_{\pi^*}}[r(s, a)] + M \leq \mathcal{J}(\pi^\#) \leq \mathbb{E}_{(s,a) \sim \rho_{\pi^*}}[r(s, a)] + M. \quad (24)$$

So we have $\mathcal{J}(\pi^\#) = \mathbb{E}_{(s,a) \sim \rho_{\pi^*}}[r(s, a)] + M$. It is worth noticing that the necessary condition for $\mathcal{J}(\pi^\#)$ to reach the upper bound, is that $\mathbb{E}_{(s,a) \sim \rho_{\pi^\#}}[r(s, a)] = \mathbb{E}_{(s,a) \sim \rho_{\pi^*}}[r(s, a)]$. Therefore, $\mathbb{E}_{(s,a) \sim \rho_{\pi^\#}}[r(s, a)] \geq \mathbb{E}_{(s,a) \sim \rho_\pi}[r(s, a)], \forall \pi \in \Pi$. That is, $\pi^\#$ is the optimal policy of original RL objective.

Based on the above two proofs, we prove the optimal policy invariance of our proposed objective. $\square$

## C    PROOFS

### C.1    LEMMA 1

**Lemma 1.** *(Demonstration Policy Evaluation). Consider the demonstration Bellman backup operator $\mathcal{T}^\pi$ in Eq. 6, and a initial Q function $Q^0 : \mathcal{S} \times \mathcal{A} \to \mathbb{R}$ with $|\mathcal{A}| < \infty$, and define $Q^{k+1} = \mathcal{T}^\pi Q^k$. Then the sequence $Q^k$ will converge to the demonstration Q-value of $\pi$ as $k \to \infty$.*

*Proof.* Given Assumption 1, we first prove that $D_{KL}(\pi, \pi_E)$ is bounded for any $\pi \in \Pi$:

$$
\begin{aligned}
&||D_{KL}(\pi, \pi_E)|| \\
&= ||\sum_a \pi(a|s) \log \frac{\pi(a|s)}{\pi_E(a|s)}|| \\
&= ||\sum_a \pi(a|s) \log \pi(a|s) - \sum_a \pi(a|s) \log \pi_E(a|s)|| \\
&\leq ||\sum_a \pi(a|s) \log \pi(a|s)|| + ||\sum_a \pi(a|s) \log \pi_E(a|s)||.
\end{aligned}
\tag{25}
$$

Given $|\mathcal{A}| < \infty$, we can get the first term $||\sum_a \pi(a|s) \log \pi(a|s)||$ is bounded by $\log |\mathcal{A}|$, according to the principle of maximum entropy.

Given the existence of $\delta \in (0, 1)$ satisfying that $\min_{a \in \mathcal{A}} \pi_E(a|s) \geq \delta, \forall s \in \mathcal{S}$, so we can get the second term $||\sum_a \pi(a|s) \log \pi_E(a|s)||$ is bounded by $\log \frac{1}{\delta}$.

Therefore, we have that $||D_{KL}(\pi, \pi_E)|| \leq \log |\mathcal{A}| + \log \frac{1}{\delta}$. We introduce $M \triangleq \log |\mathcal{A}| + \log \frac{1}{\delta}$ to represent the upper bound of $D_{KL}(\pi, \pi_E)$.

We use the notion $\bar{r}_\pi(s, a)$ to denote the reshaped reward: $\bar{r}_\pi(s, a) \triangleq r(s, a) + \mathbb{1}_{s \in \text{supp } \rho_E(s)} \cdot (M - D_{KL}(\pi(\cdot|s), \pi_E(\cdot|s)))$. Then we prove that $\bar{r}_\pi(s, a)$ is also bounded:

$$
\begin{aligned}
||\bar{r}_\pi(s, a)|| &= ||r(s, a) + \mathbb{1}_{s \in \text{supp } \rho_E(s)} \cdot (M - D_{KL}(\pi(\cdot|s), \pi_E(\cdot|s)))|| \\
&\leq ||r(s, a)|| + ||M - D_{KL}(\pi(\cdot|s), \pi_E(\cdot|s))|| \\
&\leq ||r(s, a)|| + M.
\end{aligned}
\tag{26}
$$

Supposing the extrinsic reward from the environment is bounded, the shaped reward will be also bounded from the above inequality. We continue rewriting the update rule as

$$
Q(s, a) \leftarrow \bar{r}_\pi(s, a) + \gamma \mathbb{E}_{s', a' \sim p(\cdot|s, a), \pi(\cdot|s')}[Q(s', a')].
\tag{27}
$$

Then, we can find that this bellman backup operator over $Q$ can be viewed as a special case of the standard bellman backup operator over $Q$, by instantiating the reward function $r(s, a)$ by the reshaped reward $\bar{r}_\pi(s, a)$. After applying the standard convergence results for policy evaluation (Sutton & Barto, 1998), the convergence of demonstration $Q$-value is eventually proved.    $\square$

### C.2    LEMMA 2

**Lemma 2.** *(Demonstration Policy Improvement). Let $\pi_{old} \in \Pi$ and let $\pi_{new}$ be the optimizer of the maximization problem defined in Eq. 8. Then $Q^{new}(s, a) \geq Q^{old}(s, a)$, $\forall (s, a) \in \mathcal{S} \times \mathcal{A}$ with $|\mathcal{A}| < \infty$.*

*Proof.* Let $\pi_{old} \in \Pi$ and let $Q^{\pi_{old}}$ and $V^{\pi_{old}}$ be the corresponding demonstration Q-value function and demonstration value function, and let $\pi_{new}$ be defined as

$$
\begin{aligned}
\pi_{new}(\cdot|s) &= \arg\max_{\pi' \in \Pi} \mathbb{E}_{a \sim \pi'(\cdot|s)}[Q^{\pi_{old}}(s, a)] - \mathbb{1}_{s \in \text{supp } \rho_E(s)} \cdot D_{KL}(\pi'(\cdot|s), \pi_E(\cdot|s)) \\
&= \arg\max_{\pi' \in \Pi} F_{\pi_{old}}(\pi'(\cdot|s)).
\end{aligned}
\tag{28}
$$

Given the maximum of $\pi_{new}$, we can derive that $F_{\pi_{old}}(\pi_{new}) \geq F_{\pi_{old}}(\pi_{old})$. Hence[9],

$$
\begin{aligned}
&\mathbb{E}_{a\sim\pi_{new}(\cdot|s)}[Q^{\pi_{old}}(s,a)] + \mathbb{1}_{s\in\text{supp}\,\rho_E(s)} \cdot (M - D_{KL}(\pi_{new}(\cdot|s), \pi_E(\cdot|s))) \\
&\geq \mathbb{E}_{a\sim\pi_{old}(\cdot|s)}[Q^{\pi_{old}}(s,a)] + \mathbb{1}_{s\in\text{supp}\,\rho_E(s)} \cdot (M - D_{KL}(\pi_{old}(\cdot|s), \pi_E(\cdot|s))) = V^{\pi_{old}}(s).
\end{aligned}
\tag{29}
$$

Next, we consider the demonstration bellman equation (i.e. Eq. 21):

$$
\begin{aligned}
Q^{\pi_{old}}(s,a) &= r(s,a) + \gamma\mathbb{E}_{s'\sim p(\cdot|s,a)}[V^{\pi_{old}}(s')] \\
&\leq r(s,a) + \gamma\mathbb{E}_{s'\sim p(\cdot|s,a)}[\mathbb{E}_{a'\sim\pi_{new}(\cdot|s')}[Q^{\pi_{old}}(s',a')] + \mathbb{1}_{s'\in\text{supp}\,\rho_E(s)} \cdot (M - D_{KL}(\pi_{new}(\cdot|s'), \pi_E(\cdot|s')))] \\
&\quad\vdots \\
&\leq Q^{\pi_{new}}(s,a),
\end{aligned}
\tag{30}
$$

where we have repeatedly expanded $Q^{\pi_{old}}$ on the RHS by applying the demonstration bellman equation (i.e. Eq. 21) and the inequality in Eq. 29. □

## C.3 THEOREM 1

**Theorem 1.** (Demonstration Policy Iteration). *Repeated application of demonstration policy evaluation and demonstration policy improvement from any $\pi \in \Pi$ converges to a policy $\pi^*$ such that $Q^{\pi^*}(s,a) \geq Q^{\pi}(s,a)$ for all $\pi \in \Pi$ and $(s,a) \in \mathcal{S} \times \mathcal{A}$, assuming $|\mathcal{A}| < \infty$.*

*Proof.* Let $\pi_i$ be the policy at iteration $i$. By Lemma 2, the sequence $Q^{\pi_i}$ is monotonically increasing. Since $Q^{\pi}$ is bounded above for $\pi \in \Pi$ (the shaped reward $\bar{r}_{\pi}(s,a)$ is bounded), the sequence converges to some $\pi^*$. We will still need to show that $\pi^*$ is indeed optimal. At convergence, it must be case that $F_{\pi^*(\cdot|s)}(\pi^*(\cdot|s)) \geq F_{\pi^*(\cdot|s)}(\pi(\cdot|s))$, for all $\pi \in \Pi, \pi \neq \pi^*$. Using the same iterative argument as in the proof of Lemma 2, we get $Q^{\pi^*}(s,a) \geq Q^{\pi}(s,a), \forall(s,a) \in \mathcal{S} \times \mathcal{A}$. That is, the demonstration Q-value of any other policy is lower than that of the converged policy. Hence $\pi^*$ is optimal in $\Pi$. □

## C.4 THEOREM 2

**Theorem 2.** *Given any policy $\pi$ and expert policy $\pi_E$, the optimal discriminator $D^*(s,a) = \arg\max_{D(s,a)}[\mathbb{E}_{a\sim\pi(\cdot|s)}[\log D(s,a)] + \mathbb{E}_{a\sim\pi_E(\cdot|s)}[\log(1 - D(s,a))]]$, so $\frac{\pi(a|s)}{\pi_E(a|s)} = \frac{D^*(s,a)}{1-D^*(s,a)}$.*

*Proof.* The criterion for training the discriminator $D$ is to maximize the quantity $V(D)$:

$$
\begin{aligned}
V(D) &= \int_a \pi(a|s)\log(D(s,a))da + \int_a \pi_E(a|s)\log(1 - D(s,a))da \\
&= \int_a \pi(a|s)\log(D(s,a)) + \pi_E(a|s)\log(1 - D(s,a))da.
\end{aligned}
\tag{31}
$$

For any $(a,b) \in \mathbb{R}^2 - \{0,0\}$, the function $y \to a\log(y) + b\log(1-y)$ achieves its maximum in $[0,1]$ at $\frac{a}{a+b}$. Therefore, we can infer that $D^*(s,a) = \frac{\pi(a|s)}{\pi(a|s)+\pi_E(a|s)}$. We further rewrite it in an equivalent form $\frac{\pi(a|s)}{\pi_E(a|s)} = \frac{D^*(s,a)}{1-D^*(s,a)}$. Note that this proof can be seen as the extension of Proposition 1 of GANs(Goodfellow et al., 2014) in our context. □

## D IMPLEMENTATION DETAILS

The hyperparameter setting in the experiment is provided in Table 1. We use the GAIL code[10] (implemented by Theano) released by the authors of the original GAIL paper as our GAIL baseline.

---

[9]The sense of inequality is not changed when the same number (namely, $\mathbb{1}_{s\in\text{supp}\,\rho_E(s)} \cdot M$) is added to both sides of inequality,

[10]https://github.com/openai/imitation

We also use the SAC code[11] (implemented by Tensorflow) released by the authors of the original SAC paper as our SAC baseline. The authors of POfD and DDPGfD have not publicly released their code by now. Following the similar setting in (Kang et al., 2018; Vecerik et al., 2017), we implement POfD and DDPGfD based on OpenAI Baseline[12] as our baselines. For the seeds in the evaluation, we uniformly choose the values of {0, 1, 2, 3, 4, 5, 6, 7, 8, 9} and share them in all tasks we evaluated.

| Environment | $\alpha$ | $\mu$ | $M$ | batch size | demonstration quality |
|---|---|---|---|---|---|
| Hopper-v1 | 2 | 100 | 1.5 | 256 | 2648 |
| HalfCheetah-v1 | 1 | 10 | 1.5 | 256 | 4478 |
| Walker2d-v1 | 0.5 | 10 | 1.5 | 256 | 2431 |
| Ant-v1 | 0.5 | 100 | 1.5 | 256 | 1923 |
| Humanoid-v1 | 0.5 | 1 | 1.5 | 256 | 4647 |

Table 1: Hyper-parameters in our DAC implementation, where $\alpha$ denotes the extrinsic reward scale coefficient, $\mu$ stands for the temperature coefficient, $M$ denotes the practical upper bound of augmented reward, batch size denotes the sample number of each policy update, and demonstration quality denotes the mean value of demonstration trajectories returns.

The original hyper setting in the GAIL code is not that sample efficient. Following the practical suggestions from (Jeon et al., 2018), we reduce the batch size of state-action pairs sampled in each iteration from 50000 to 1000 for *Hopper-v1, Walker2d-v1, HalfCheetah-v1,* and from 50000 to 5000 for *Ant-v1* and *Humanoid-v1*, to improve the sample efficiency of GAIL algorithm.

Besides, in our detailed implementation, we adopt the simple clip technique to ensure that the augmented reward is bounded in $(0, M)$, which keeps up with our derived theory and makes the training process more stable and effective.

---

[11] https://github.com/haarnoja/sac
[12] https://github.com/openai/baselines

