# OpenReview forum: "Demonstration Actor Critic"
_ICLR.cc/2020/Conference — Reject_

### Official Review · AnonReviewer3 · 2019-10-09
**Official Blind Review #3**

**Rating:** 6

**Review:**

Summary:
This paper studied reinforcement learning from demonstration. Given a set of
expert demonstrations, this work provides a policy-dependent reward shaping objective that
can utilize demonstration information and preserves policy optimality, policy improvement,
and the convergence of policy iteration at the same time, under the assumption
that expert policy is optimal and stochastic.
The main advantage of the proposed method is that the reward shaping function
is related to the current policy.
A practical algorithm based on theoretical derivation is provided. The authors conducted sufficient experimental results to demonstrate that the proposed method is effective,
comparing with a set of advanced baselines.

I recommend acceptance:
Previous works on RLfD usually empirically incorporated a regularization
to the RL objective, while those works didn't discuss whether this
regularization will lead to sub-optimal policy or not. This paper discussed
how to use the demonstration information to do exploration and maintain
policy invariance at the same time, with a relatively strong assumption.
Using the framework from SAC, the
algorithm is shown to converge to the optimal via policy iteration, in
tabular case. This work also developed a practical expert policy
support estimation algorithm to measure the uncertainty of
expert policy. Utilizing the adversarial training framework,
the explicit computation of expert policy is avoided.
The authors conducted sufficient experiments to demonstrate
the effectiveness of the proposed method, compared with the state-of-the-art in RLfD.


Technical concerns:
The stochasticity assumption of expert policy in Asm. 1 can be contradicted
with that expert policy is optimal in policy invariance proof.
This paper works on a problem of infinite horizon discounted MDP.
According to Puterman [1994], Theorem 6.2.7, there always exists a
deterministic stationary policy \pi that is optimal. Or intuitively,
if we find the optimal value function via Bellman optimality equation,
the optimal policy is acting greedily (deterministic). The provided theorems are
not compatible with the MDP where only deterministic optimal policy exists.
It is not clear that in what type of MDPs the optimal stochastic policy exists and
it can satisfy Asm. 1.
Could the authors clearly specify the applicable problem settings?

If the asm 1 is satisfied, what is the necessity to incorporate the
indicator function in Eq 4.? Since p(s) > 0 for all s, following strong stochasticity policy.
For any trajectory \tau, p(\tau) = p(s)\Pi_t p(s_{t+1}|s_{t}, a_t)\pi_E(a_t|s_t) > 0.

The proof of Theorem 2 is similar to the proof in Proposition 1, [1], though in a
different context. It would be better to have a citation?

Experiments:
It would be more convincing to show the performance of behavior cloning policy using
expert trajectories.

[1] Goodfellow, Ian, et al. "Generative adversarial nets." Advances in neural information processing systems. 2014.



**Experience Assessment:**

I have read many papers in this area.

**Review Assessment: Checking Correctness Of Derivations And Theory:**

I carefully checked the derivations and theory.

**Review Assessment: Checking Correctness Of Experiments:**

I assessed the sensibility of the experiments.

**Review Assessment: Thoroughness In Paper Reading:**

I read the paper at least twice and used my best judgement in assessing the paper.

---

> ### Author Response · Authors · 2019-11-11
> **Response to AnonReviewer3**
>
> Thank you so much for your supportive and constructive comments! we will give response to every question you have. Please let us know if you have any question on our response.
>
> Summary: our response includes: (1) Discussion on stochasticity assumption. (2) Proof of theorem 2 (3) Performance of behavior cloning policy.
>
> First of all, we really appreciate your affirmation of our efforts to investigate the optimal policy invariance and others!
>
> ** Discussion on stochasticity assumption **
> Thanks a lot for your careful and insightful thought here. The provided theorems are indeed only compatible with the MDP where the optimal stochastic policy exists and satisfy Asm. 1 now, due to the stochasticity assumption. To better generalize our theory part into more general MDPs (e.g. only deterministic optimal policy exists) you mentioned,  one potential answer we provided is to replace the KL divergence with some naturally-bounded divergences (e.g. Jensen-Shannon divergence, satisfying that $0 \le JSD(p ||q) \le 1$ given the base 2 logarithm), since essentially the primary purpose of stochasticity assumption is to ensure our policy-dependent shaping reward, which includes KL divergence term, is bounded. In this case, we will no longer need to introduce stochasticity assumption, and the theory part can be more close to the practical cases (e.g. only deterministic optimal policy exists). Considering some potential further modifications to policy evaluation/policy improvement parts (from KL towards JS), we leave this into our future work.
>
> It’s true that if the Asm. 1 is satisfied, the indicator function is unnecessary. But empirically, the indicator function is very useful, especially when demonstration states often only cover a part of the whole state space. Intuitively speaking, such indicator function aims to encourage the agent to visit the demonstrated states  (states visited by the expert strategy).
>
> ** Proof of theorem 2 **
> Thanks for your kind reminder, the proof of our theorem 2 is indeed inspired by that of Proposition 1 in Generative Adversarial Nets (GANs). We cited the GAN paper above the theorem 2 in Section 4.2, and we are willing to refer to it again in our proof part for clearer illustration (presented in the C.4 Theorem 2 part in Appendix).
>
> ** Performance of behavior cloning policy **
> Thanks for your advice, and we will also add the performance of behavior cloning policy using expert trajectories in our camera-ready version.
>
> We welcome further discussion and are willing to answer any further questions. Thanks for your time and valuable feedbacks.

---

### Official Review · AnonReviewer1 · 2019-10-22
**Official Blind Review #1**

**Rating:** 6

**Review:**

This paper presents a method for doing RL from demonstrations in continuous control tasks. It combines both an augmented reward for minimizing the KL between the policy and the expert actions as well as directly minimizing that KL in the policy. Results on 5 sparse reward mujoco tasks show that it out-performs other related methods.


The motivation for the paper is difficult to follow. They claim that using demonstration data in a supervised manner "cannot generalize supervision signal over those states unseen in the demonstrations," but most of these approaches are using neural networks and definitely are generalizing those signals to other states. Whether they're generalizing accurately or not is a different question. In contrast, they say that reward shaping approaches do not suffer that problem because they evaluate trajectories rather than states, but there will still be a problem of generalizing to new trajectories.

The abstract is even more confusing as it tries to jump straight into the issues with these approaches without any explanation. I don't think there's enough space in the abstract to go into that level of detail.

The description of DQfD and DDPGfD in the related work is not accurate. They're described as "treating demonstration data as self-generated data," but in fact they both add supervised losses to more closely match the demonstrated data. https://ieeexplore.ieee.org/document/8794074 is another method built on DDPG that has both a critic and actor loss like yours and would make a useful comparison.

The related work section should also discuss and compare/contrast to GAIL, I was surprised that wasn't in there, especially since you also use a discriminator to differentiate expert and agent actions.

The end of the related work section is not very clear, you say these methods are problematic because "the adopted shaping reward yields no direct dependence on the current policy" but there's no explanation or motivation for why that would be a problem.

Assumption 1 seems like a very strong assumption that would not be true for many human experts.

For the experiments, I wonder about the impact of only using sparse reward tasks. Converting the tasks to sparse reward in this way makes them partially observable, and then potentially the expert demonstrations are required to overcome that partial observability. How do the methods compare on the unmodified tasks? There was nothing specific in your algorithm that meant it should specifically address sparse reward tasks. What about tasks that are naturally sparse reward?

Overall, the algorithm is interesting and the results are nice. The motivation and related work need to be made clearer to situate this work with the other related works. And the experiments should go beyond these tasks that have been modified to have sparse rewards.


The revised version of the paper addresses many of my concerns about the motivation, related works, and comparisons with GAIL, so I'm updating my score to Weak Accept.

**Experience Assessment:**

I have published one or two papers in this area.

**Review Assessment: Checking Correctness Of Derivations And Theory:**

I assessed the sensibility of the derivations and theory.

**Review Assessment: Checking Correctness Of Experiments:**

I carefully checked the experiments.

**Review Assessment: Thoroughness In Paper Reading:**

I read the paper at least twice and used my best judgement in assessing the paper.

---

> ### Author Response · Authors · 2019-11-11
> **Response to AnonReviewer1[1/2]**
>
> Thanks so much for your comments and suggestions! In the following context, we will give response to every question you have. Please let us know if you have any question on our response.
>
> Summary: our response includes: (1) Clarification on our motivation; (2) Clarification on related works; (3) Clarification on our assumption; (3) Explanations of experiment environment.
>
> ** Clarification on our motivation **
> We are sorry that we should give a clearer description of the motivation of our work. We give a new description here:
>
> We study the problem of RLfD, where both reward signals and expert demonstrations are given. One approach leverages demonstration data in a supervised manner. Though simple and direct, such approach can only provide accurate supervision signal over those states seen in the demonstrations [1, 2, 5]. To address this issue, another approach uses demonstration data for reward shaping, and can provide guidance on how to take actions, even for those states are not seen in the demonstrations. Specifically, such reward shaping approach trains an agent not only to imitate demonstrated actions when it encounters demonstrated states, but also to reach demonstrates states (states visited by the expert strategy), when it confronts states that are not observed in the demonstration data [3, 4, 5]. This is the core idea behind such reshaping reward based approach. However, since the new adopted shaping reward yields no direct dependence on the current policy, such approach, updating policy over demonstrated states in the same way as other states by the reshaped value function, overlook the validity of such direct supervision for demonstrated states during training.
>
> In this paper, we delicately design a policy dependent shaping reward: $1_{s \in supp{\rho_{\pi_E}(s)}} \cdot (M – D_{KL}(\pi_{\theta}(\cdot|s),\pi_{E}(\cdot|s)))$, in order to provide both guidance for all states, as well as direct supervision for demonstrated states. The purpose of designing the indicator function of $supp{\rho_{\pi_{E}(s)}}$ and $M$ is to assign positive reward for demonstrated states, while assign zero reward for non-demonstrated states, which will encourage the agent to reach demonstrated states. And the purpose of designing $D_{KL}(\pi_{\theta}(\cdot|s),\pi_{E}(\cdot|s))$ is to encourage the agent to directly imitate the demonstrated actions over these demonstrated states. In particular, optimizing the objective with such policy dependent shaping reward will directly lead to a new update rule, as shown in Eqn. 8 in Section 4.1, consisting two key parts: 1) encourage the agent to obtain more cumulative rewards and reach the demonstrated states as much as possible. 2) determine whether current state belongs to demonstrated states by the indicator function $1_{s \in supp{\rho_{\pi_E}(s)}}$, if so, leverage the direct supervision signals over current demonstrated state.
>
> Following your suggestion, we have updated the introduction and abstract section in our paper.

---

> ### Author Response · Authors · 2019-11-11
> **Response to AnonReviewer1[2/2]**
>
> ** Clarification on related works **
> Thanks for pointing out our mistake on stating DQfD. We have updated the statement in our updated paper (presented in the second paragraph of Section 2). Besides, we also add reference of the relevant paper (https://ieeexplore.ieee.org/document/8794074 ) you mentioned in our paper.
>
> Regarding GAIL method, thanks for your suggestions! We have added the discussion/comparison with GAIL in our related work section, and empirically we also compared GAIL in our experiment section.
>
> ** Clarification on our assumption **
> We agree that the assumption 1 is a relatively strong assumption for many human experts. The main purpose of this assumption is to theoretically ensure the policy-dependent KL shaped reward is bounded, which can further lead to the convergence of demonstration policy evaluation. In practice, our method can also empirically work well for the deterministic expert situations by reward clipping technique, as shown in Figure 1 in section 5.1.
>
> ** Explanations of experiment environment **
> Recent RLfD methods tend to demonstrate their effectiveness on the sparse environments [4, 6, 7], where the demonstration data may have more impact on aiding exploration. Following this line, we adopt the delayed version of sparse MuJoCo (publicly available in https://github.com/Hwhitetooth/lirpg), as widely used in (Zheng et al., 2018; Oh et al., 2018; Guo et al., 2019). Thanks for your advice, and we are very willing to add more experiments on the tasks that are naturally sparse in our camera-ready version.
>
> We hope our rebuttal and paper revision could address your concerns. We welcome further discussion and are willing to answer any further questions.
>
> [1] Tim Brys et al. “Reinforcement Learning from demonstration through shaping.” In IJCAI 2015.
> [2] Aravind Rajeswaran et al. “Learning complex dexterous manipulation with deep reinforcement learning and demonstrations.” In RSS 2018.
> [3] Jonathan Ho and Stefano Ermon. “Generative adversarial imitation learning.” Advances in Neural Information Processing System. 2017.
> [4] Bingyi Kang, et al. “Policy optimization with demonstrations.” International Conference on Machine Learning. 2018.
> [5] Siddharth Reddy, et al. “SQIL: imitation learning via regularized behavior cloning.” arXiv: 1905.11108.
> [6] Ashvin Nair, et al. “Overcoming exploration in reinforcement learning with demonstrations.” arXiv:1709.10089.
> [7] Mel Vecerik, et al. “Leveraging demonstrations for deep reinforcement learning on robotics problems with sparse rewards.” arXiv: 1707.08817.

---

> > ### Comment · AnonReviewer1 · 2019-11-14
> > **Response**
> >
> > Thanks for the response and the paper revisions. I do think the paper is much stronger now with the revised introduction, abstract, and related works, and the comparison with GAIL in the experiments. I will update my review score to a 6.

---

### Official Review · AnonReviewer2 · 2019-10-24
**Official Blind Review #2**

**Rating:** 1

**Review:**

This paper proposes to mix reinforcement learning and imitation learning to boost the learning of an actor critic architecture. The authors use a regularized reward function that minimizes the divergence between the policy of the expert and the one followed by the agent. They use demonstrations obtained from a trained agent and experiment their method on several mujoco tasks.

I have many concerns about this paper. First, The state of the art is missing important pre-deep-learning references such as:

1. Direct Policy Iteration with Demonstrations: Chemali and Lazaric
2. Learning from limited demonstrations, Beomjoon et al.
3. Residual Minimization Handling Expert Demonstrations, Piot et al.

Then, they make a mistake by saying that DQfD only considers transitions from the expert as self-generated and placed in the replay buffer. DQfD actually uses the same additional structured classification loss than Piot et al. [3] (except that they use boosted trees instead of deep networks, DQfD and Piot et al. are the same algorithm).

Also, the proposed solution here is equivalent to regularizing the MDP with a KL divergence  w.r.t. to an initial policy that would be the one of the expert. It is already studied in several works and more generally it comes with some assumptions on the policy update. It is generally studied in

4. A theory of regularized MDP, Geist et al

They actually propose exactly the same framework as a special case in the appendix of that paper.

In addition to not be very novel, I think the method has some flaws. The authors use demonstrations coming from a pre-trained network which is known to make the imitation learning part much easier. Especially if it comes from an RL agent using similar deep RL algorithms (which is the case here). Finally, they only test on mujoco tasks which are very specific tasks with deterministic dynamics and very dense rewards  around states visited by the optimal strategy so initializing with an expert policy that is learned from demonstrations of a similar network of course helps. I would be more impressed by experiments on stochastic environments and sparse rewards.

Finally, there is a concurrent work submitted to the same conference. Of course the authors could not know but I’d like to have their impression about how their work is different.

https://openreview.net/forum?id=BJg9hTNKPH&noteId=BJg9hTNKPH

**Experience Assessment:**

I have published in this field for several years.

**Review Assessment: Checking Correctness Of Derivations And Theory:**

I carefully checked the derivations and theory.

**Review Assessment: Checking Correctness Of Experiments:**

I carefully checked the experiments.

**Review Assessment: Thoroughness In Paper Reading:**

I read the paper thoroughly.

---

> ### Author Response · Authors · 2019-11-11
> **Response to AnonReviewer2 [1/2]**
>
> Thanks so much for your review and detailed comments. In the following parts, we will give responses for each concern you have. Please let us know if you have any question on our response.
>
> Our response generally includes: (1) Missing important pre-deep-learning references; (2) Clarification on DQfD method; (3) The concern of novelty; (4) Explanation of experiment details; (5) Discussion on the concurrent work BRAC.
>
> ** Missing important pre-deep-learning references **
> We are sorry that we missed some important pre-deep-learning works you mentioned. These methods are indeed relevant and we have added reference to them in our updated paper (presented in the second paragraph of Section 2).
>
> ** Clarification on DQfD method **
> We apologize for our mistake in DQfD part. We have updated the previous statement, as shown in our updated paper (presented in the second paragraph of Section 2). Thanks for pointing it out!
>
> ** The concern of novelty **
> To clarify the novelty of our work, we start with our motivation:
> 1. We study the problem of RLfD, where both reward signals and expert demonstrations are given. One approach leverages demonstration data in a supervised manner. Though simple and direct, such approach can only provide supervision signal over those states seen in the demonstrations. To address this issue, another approach uses demonstration data for reward shaping. Such approach trains an agent not only to imitate demonstrated actions over these demonstrated states, but also to reach the demonstrated states (states visited by the expert strategy), when it confronts states that are unseen in the demonstration data [1, 2, 3]. This is the core idea behind such reshaping reward based approach. However, since the new adopted shaping reward yields no direct dependency on the current policy, such approach, updating policy over demonstrated states in the same way as other states by the reshaped value function, overlooks the validity of such direct supervision for demonstrated states during training.
>
> 2. In this paper, we delicately design a policy dependent shaping reward: $1_{s \in supp{\rho_{\pi_E}(s)}} \cdot (M – D_{KL}(\pi_{\theta}(\cdot|s),\pi_{E}(\cdot|s)))$, in order to provide both guidance for all states, as well as direct supervision for demonstrated states. The purpose of designing the indicator function of $supp{\rho_{\pi_{E}(s)}}$ and $M$ is to assign positive reward for demonstrated states, while assign zero reward for non-demonstrated states, which will encourage the agent to reach demonstrated states. And the purpose of designing $D_{KL}(\pi_{\theta}(\cdot|s),\pi_{E}(\cdot|s))$ is to encourage the agent to directly imitate the demonstrated actions over these demonstrated states. In particular, optimizing the objective with such policy dependent shaping reward will directly lead to a new update rule, as shown in Eqn. 8 in Section 4.1, consisting two key parts: 1) encourage the agent to obtain more cumulative rewards and reach the demonstrated states as much as possible. 2) determine whether current state belongs to demonstrated states by the indicator function $1_{s \in supp{\rho_{\pi_E}(s)}}$, if so, leverage the direct supervision signals over current demonstrated state.
>
> By contrast, although very general, these Regularized MDP methods only regularize MDP with a KL divergence term, and are not very suitable for the RLfD problem. These methods do not encourage the agent to reach demonstrated states (states visited by the expert strategy) explicitly, but it is a very important unique property of RLfD problem itself [1, 2, 3]. Besides, most of Regularizing MDP works assume that the explicit expression of initial policy is available, but in our case, we can only access to expert demonstrations. To this end, we use the GAN technique to avoid the usage of explicit expression of $\pi_{E}$, and take advantage of support estimation techniques to estimate the indicator function of $supp{\rho_{\pi_{E}(s)}}$ from expert demonstrations, which also leads to another obvious difference.
>
> Although there exists a significant difference between the scenario of our work and that of some Regularized MDP works, our method is indeed formally similar to these works. And we are very willing to add these Regularizing MDP works into our paper, and discuss their relationship and difference with our method (please refer to the last paragraph of Section 2).

---

> > ### Comment · AnonReviewer2 · 2019-11-14
> > **Answer to authors**
> >
> > The approach of regularized MDPs and BRAC works can also be seen as a "reward shaping" (they modify the reward by adding a divergence term). And it is shown that they don't lead to the optimal policy for the original MDP at the end of the day (but not too far). The difference with this work is that they don't try to approximate the indicator function.
> >
> > I'm actually concerned about learning the indicator function. It basically consists in learning a harsh approximation of the density of visits of the expert for every state which is a quite hard task, especially in stochastic environments. So, again, I think the choice of the test environment is very important to demonstrate the performance of the algorithm. It seems really designed to solve exactly the tasks that are chosen in the experimental section which is not so convincing to me.

---

> > > ### Author Response · Authors · 2019-11-15
> > > **Thank you for your reply**
> > >
> > > Thank you for your reply, and we will continue giving responses for each concern you have.
> > >
> > > Our response generally includes: (1) Regarding the approach of regularized MDPs; (2) Regarding indicator function learning and choice of test environment.
> > >
> > > ** Regarding the approach of regularized MDPs **
> > > We agree that the approach of regularized MDPs is very general and can also be seen as one kind of “reward shaping”. However, these methods only modify the reward with a KL divergence term, and are not really very suitable for the RLfD problem we studied. Since these methods do not encourage the agent to reach demonstrated states (states visited by the expert strategy) explicitly, but remarkably, it is a very important unique property of RLfD problem itself, as widely discussed in [1, 2, 4].
> > >
> > > In contrast, we propose to delicately design a new shaping reward: $1_{s \in supp{\rho_{\pi_E}(s)}} \cdot (M – D_{KL}(\pi_{\theta}(\cdot|s),\pi_{E}(\cdot|s)))$, specifically for the RLfD scenario. The purpose of designing the indicator function of $supp{\rho_{\pi_{E}(s)}}$ and $M$ is to assign positive reward for demonstrated states, while assign zero reward for non-demonstrated states, which will encourage the agent to reach demonstrated states. And the purpose of designing $D_{KL}(\pi_{\theta}(\cdot|s),\pi_{E}(\cdot|s))$ is to encourage the agent to directly imitate the demonstrated actions over these demonstrated states.
> > >
> > > Given such delicately designed shaping reward, we also strictly prove the optimal policy invariance of corresponding objective function, under the assumption that the expert policy $\pi_{E}$ is the optimal policy (please refer to Appendix B for detailed proof). In contrast, it is shown that these regularized MDPs methods may not lead to the optimal policy for the original MDP.
> > >
> > > ** Regarding indicator function learning and choice of test environment **
> > > Regarding indicator function learning, we agree that ideally, computing the indicator function of $supp\rho_{\pi_E}(s)$ w.r.t every state is a quite hard task. However, in practice, we are given only a finite number of expert demonstrations sampled by $\pi_{E}$, and the practical estimation from the limited number of demonstrations states will not that hard (we admit that there may exist some of estimation error here, due to the inevitable finite-sample estimation, and the similar case has also been discussed in [3]). Please note that no matter whether the environment is stochastic or deterministic, we all estimate the practical indicator function from the limited number of demonstrations states, so these two cases will be not very different.
> > >
> > > In order to practically estimate the indicator function of $supp\rho_{\pi_E}(s)$, we consider taking advantage of support estimation techniques. Recently, RED [3] have established a connection between support estimation ideas and Random Network Distillation (RND). They estimate $1_{(s,a) \in supp{\rho_{\pi_E}(s, a)}}$ by using prediction errors of networks trained on the expert state-action pairs, and use the estimation as the recovered reward function for policy training, in the context of Imitation Learning. Inspired by their work, we similarly use the prediction errors of networks trained on demonstrated states to estimate $1_{s \in supp{\rho_{\pi_E}(s)}}$, as our indicator function. We think both the experiment results of their work and ours can well support the effectiveness of such indicator function estimation. Note that RED also conducts their experiments in MuJoCo tasks (in Section 4.2 of the paper).
> > >
> > > We agree that the choice of test environment is very important, and here we will explain how we choose the environment. Recent RLfD methods tend to demonstrate their effectiveness on the sparse environments [4, 5, 6], where the demonstration data may have more impact on aiding exploration. It is worth mentioning that the most related work POfD also conducts experiments in MuJoCo after sparsifying (in Section 6 of the paper). Following this line, we adopt the delayed version of sparse MuJoCo (publicly available in https://github.com/Hwhitetooth/lirpg ), as widely used in previous works (Zheng et al., 2018; Oh et al., 2018; Guo et al., 2019).
> > >
> > > We really hope the above explanations could address your concerns. Thanks for your time and valuable feedbacks again.
> > >
> > > [1] Jonathan Ho and Stefano Ermon. “Generative adversarial imitation learning.” NIPS 2017.
> > > [2] Siddharth Reddy, et al. “SQIL: imitation learning via regularized behavior cloning.” arXiv:1905.11108.
> > > [3] Ruohan Wang, et al. “Random Expert Distillation: Imitation Learning via Expert Policy Support Estimation.” ICML 2019.
> > > [4] Bingyi Kang, et al. “Policy optimization with demonstrations.” ICML 2018.
> > > [5] Ashvin Nair, et al. “Overcoming exploration in reinforcement learning with demonstrations.” arXiv:1709.10089.
> > > [6] Mel Vecerik, et al. “Leveraging demonstrations for deep reinforcement learning on robotics problems with sparse rewards.” arXiv:1707.08817.

---

> ### Author Response · Authors · 2019-11-11
> **Response to AnonReviewer2 [2/2]**
>
> ** Explanation of experiment details **
> Regarding demonstration details, we follow the common practice of recent works in imitation learning and RLfD [1, 2], and collect demonstration data from the expert policy released by the authors of the original GAIL [4], which can ensure the accessibility and reproducibility of these demonstrations. And for the fairness during evaluation, we used the exactly same demonstration data for all evaluated methods.
>
> Regarding test environment, we agree your opinion that for the simple tasks that, for example, only have one optimal path, initializing with expert policy from the demonstrations can apparently help. But note that some recent algorithms [5, 6] have empirically demonstrated to train a multi-modal policy in MuJoCo tasks, and actually we use the delayed version of MuJoCo tasks, which is even harder (POfD [2] also conducts experiments in MuJoCo after sparsifying). Thus, we think our experiments on current environments are convincing enough. Nevertheless, we are very willing to add experiments on more stochastic environments with sparse rewards in our camera-ready version.
>
> ** Discussion on the concurrent work BRAC **
> Thanks for mentioning this concurrent work (BRAC), and we are also very willing to discuss the relationship with this paper. Under the batch RL setting,  BRAC proposes a general framework by regularizing MDP with the divergence (e.g. MMD or KL divergence) between the learned policy and the underlying behavior policy $\pi_b$, i.e. $D(\pi(\cdot│s_t ),\pi_b (\cdot|s_t))$, which has the almost the same formulation as these regularized MDP works. In contrast, our method proposes to delicately design a new shaping reward: $1_{s \in supp{\rho_{\pi_E}(s)}} \cdot (M – D_{KL}(\pi_{\theta} || \pi_{E}))$, specifically for the RLfD scenario we studied.
>
> We hope the above explanations could address your concerns. Please kindly check our updated paper with clarification. Thanks for your time and valuable feedbacks.
>
> [1] Jonathan Ho and Stefano Ermon. “Generative adversarial imitation learning.” Advances in Neural Information Processing System. 2017.
> [2] Bingyi Kang, et al. “Policy optimization with demonstrations.” International Conference on Machine Learning. 2018.
> [3] Siddharth Reddy, et al. “SQIL: imitation learning via regularized behavior cloning.” arXiv: 1905.11108.
> [4] Public Github repository of GAIL: https://github.com/openai/imitation
> [5] Tuomas Haarnoja, et al. “Soft Actor Critic: Off-Policy Maximum Entropy Deep Reinforcement Learning with a Stochastic Actor.”  International Conference on Machine Learning. 2018.
> [6] Ziyu Wang, et al. “Robust Imitation of Diverse Behaviors.” Advances in Neural Information Processing System. 2017.

---

### Comment · Area_Chair1 · 2019-11-13
**Thanks for your reviews. Please take a look at the rebuttal.**

Dear reviewers,

Thank you very much for your efforts in reviewing this paper.

The authors have provided their rebuttal. It would be great if you take a look at them, and see whether it changes your opinion in anyway. If there is still any unclear point or a serious disagreement, please bring it up. Also if you are hoping to see a specific change or clarification in the paper before you update your score, please mention it.

The authors have only until November 15th to reply back.

I also encourage you to take a look at each others’ reviews. There might be a remark in other reviews that changes your opinion.

Thank you,
Area Chair

---

### Decision · Program_Chairs · 2019-12-19

**Decision:**

Reject

**Comment:**

The paper proposes to combine RL and Imitation Learning. It defines a regularized reward function that minimizes the KL distance between the policy and the expert action. The formulation is similar to the KL regularized MDPs, but with the difference that an additional indicator function based on the support of the expert’s distribution is multiplied to the regularized term.

Several issues have been brought up by the reviewers, including:
* Comparison with pre-deep learning literature on the combination of RL and imitation learning
* Similarity to regularized MDP framework
* Assumption 1 requiring a stochastic expert policy, contradicting the policy invariance claim
* Difficulty of learning the indicator function of the support of the expert’s data distribution

Some of these issues have been addressed, but at the end of the day, one of the expert reviewers was not convinced that the problem of learning an indicator function is going to be easy at all. The reviewer believes that learning such a function requires "learning a harsh approximation of the density of visits of the expert for every state which is a quite hard task, especially in stochastic environments.”

Another issue is related to the policy invariance under the optimal expert policy. In most MDPs, the optimal policy is not stochastic and does not satisfy Assumption 1, so the optimal policy invariance proof seems to contradict Assumption 1.

Overall, it seems that even though this might become a good paper, it requires some improvements. I encourage the authors to address the reviewers’ comments as much as possible.